# Culture Dependent and Independent Analysis of Potential Probiotic Bacterial Genera and Species Present in the Phyllosphere of Raw Eaten Produce

**DOI:** 10.3390/ijms20153661

**Published:** 2019-07-26

**Authors:** Sascha Patz, Katja Witzel, Ann-Christin Scherwinski, Silke Ruppel

**Affiliations:** 1Algorithms in Bioinformatics, ZBIT Center for Bioinformatics, University of Tübingen, Sand 14, 72076 Tübingen, Germany; 2Leibniz Institute of Vegetable and Ornamental Crops, Theodor-Echtermeyer-Weg 1, 14979 Großbeeren, Germany

**Keywords:** fresh produce, lactic acid bacteria (LAB), MALDI-TOF MS biotyping, phyllosphere microbiota, PhyloChip^®^, probiotics

## Abstract

The plant phyllosphere is colonized by a complex ecosystem of microorganisms. Leaves of raw eaten vegetables and herbs are habitats for bacteria important not only to the host plant, but also to human health when ingested via meals. The aim of the current study was to determine the presence of putative probiotic bacteria in the phyllosphere of raw eaten produce. Quantification of bifidobacteria showed that leaves of *Lepidium sativum* L., *Cichorium endivia* L., and *Thymus vulgaris* L. harbor between 10^3^ and 10^6^ DNA copies per gram fresh weight. Total cultivable bacteria in the phyllosphere of those three plant species ranged from 10^5^ to 10^8^ CFU per gram fresh weight. Specific enrichment of probiotic lactic acid bacteria from *C. endivia*, *T. vulgaris,*
*Trigonella foenum-graecum* L., *Coriandrum sativum* L., and *Petroselinum crispum* L. led to the isolation of 155 bacterial strains, which were identified as *Pediococcus pentosaceus*, *Enterococcus faecium*, and *Bacillus* species, based on their intact protein pattern. A comprehensive community analysis of the *L. sativum* leaves by PhyloChip hybridization revealed the presence of genera *Bifidobacterium*, *Lactobacillus*, and *Streptococcus*. Our results demonstrate that the phyllosphere of raw eaten produce has to be considered as a substantial source of probiotic bacteria and point to the development of vegetables and herbs with added probiotic value.

## 1. Introduction

Plants host a wide range of microorganisms, i.e., bacteria, fungi, archaea, oomycetes, and viruses, which constitute their entire microbiome. Capturing the function and complexity of the microbial community under diverse environmental impacts, the so-called phytobiome, has advanced tremendously in the last decade. Using a range of microbial model organisms, a detailed picture emerged on how microorganisms colonize plants, and how they influence the plant’s health and productivity [1]. However, microorganisms residing as plant endophytes or epiphytes usually form communities and our understanding of the influence of host genotype, host compartment, environmental factors and microbiota relationships on the phytobiome needs to be enhanced in order to use tailored microbiota for crop production [2,3].

Large efforts are being taken to illuminate the compositions and functions mainly of rhizosphere microbiota with the aim to increase nutrient uptake efficiency and resilience against soil-borne diseases in crops [4]. Compared to the rhizosphere microbiota, phyllosphere microbiota is less diverse and lower abundant [5,6]. Their ecological role, however, is equally significant for plant health and fitness and depends on the structure and function of the microbiome [7]. While community composition of rhizosphere microbiota is governed by the soil type, the major driver for phyllosphere microbiota seems to be the host genotype [8,9].

Leff and Fierer [10] analyzed the bacterial communities found on the surface of vegetables and fruits, and demonstrated that community composition and diversity is dependent on the produce type. They also identified differences between conventional and organically farmed produce, showing that organic-labeled spinach, lettuce and tomatoes had a greater OTU richness, but peaches and grapes had a lower OTU richness, when compared to the conventional produce type. Using both, a culture dependent and an independent approach, leafy vegetables (spinach and lettuce varieties) were profiled for their bacterial communities and revealed a distinct plant genotype-specific pattern [11]. Similar results were found for fungal communities that also revealed a genotypes-specific variation in lettuce phyllosphere [12]. Lopez-Velasco et al. [13] showed further that spinach leaves harbor a different composition of bacterial populations as compared to seeds or cotyledons. Due to outbreaks of foodborne illnesses caused by the consumption of vegetables, food safety and persistence of human pathogens on crops became a focus in phyllosphere microbiota research [8]. Especially raw eaten leafy vegetables and herbs are analysed to understand the effect of farming practice, postharvest processing and storage conditions of fresh produce on microbiota composition [14]. Here, a recent study compared the bacterial community composition of ready-to-eat rocket and spinach [15]. Authors demonstrated again that community composition was dependent on the plant species and that refrigeration decreased bacterial richness and led to the dominance of cold-adapted bacteria. Other studies investigated the colonization and persistence of potential human pathogenic bacteria in leaf vegetable phyllospheres [16].

In contrast, probiotic bacteria, such as strains of the lactic acid bacteria (LAB), like *Lactobacillus* spp., *Streptococcus* spp., *Enterococcus* spp. or *Pediococcus* spp., and of bifidobacteria, are recognized for their beneficial effects on the human intestinal microbiota and overall health for a long time [17,18,19,20]. Moreover, recent studies give strongly evidence on their strain- and disease-dependent probiotic efficacy [21]. The effects are addressed through the production of antimicrobial compounds, high colonization competence, immunomodulation of the host and inhibition of bacterial toxin production [22,23]. In most cases, these bacteria are used as starter cultures in the production of fermented foods, such as yoghurt, sauerkraut or rice wine [24]. Next generation probiotics have been proposed in the last decades, like strains of *Weissella* spp. or *Akkermansia muciniphila* [25,26,27]. Beside the health-promoting and antipathogenic effects of probiotics, there are reports on their risk to become a (opportunistic) pathogen, as closely related virulent species are known or because of the risk of exchanging antibiotic resistance genes [28,29,30,31,32]. The challenge here is to explore strains, like of the LAB group, especially Lactobacilli and Enterococci, which are predestined to counteract for example foodborne pathogens [33].

However, while vegetable leaves are well populated with bacteria including potential probiotic acting *Lactobacillus* strains [34,35,36,37], so far, plant-derived produce has only been considered as vehicles of probiotic cultures [38,39]. A detailed understanding of how probiotic bacteria colonize leafy vegetables or herbs would enable future approaches to enrich these tissues with the respective bacterial strains to elevate vegetable leaves beyond their basic nutritional values.

The objective of the present study was to search for bacterial species which are known to contain potential probiotic strains in the phyllosphere of raw consumed edibles and to bring them into in vitro pure cultures for further detailed studies. Therefore, we started with the quantification of the specific bifidobacterial gene copy numbers in raw eaten produce. The investigated plant species were selected due to their widespread raw consumption, economic value, and their affiliation to different plant orders, assuming that the plant genotype affects the bacterial community composition. Based on the results we then used culture-dependent approaches and assessed the plant species-specific composition of LAB. Finally, the cress phyllosphere microbiome was investigated comprehensively by applying the PhyloChip G3 technology. Altogether, we confirm that the bacterial composition of the phyllosphere of raw eaten edibles is highly dependent on the host genotype and we show that leaves harbor a wide range of potential probiotic acting bacterial genera and species.

## 2. Results

### 2.1. Quantification of Potential Probiotic Bifidobacteria in the Phyllosphere of Edible Plants

*Lepidium sativum*, family Brassicaceae, and plants belonging to two other families—*Cichorium endivia* (Asteraceae) and *Thymus vulgaris* (Lamiaceae)—were grown under greenhouse conditions in two independent experiments to collect phyllosphere samples. The aim was to search, in a culture independent approach, for the presence of bifidobacteria and to quantify their specific gene copy numbers. Using quantitative real-time PCR and *Bifidobacterium* genera-specific primers [40], between 10^3^ to 10^6^ copies per gram plant fresh weight were detected in all plant samples (Figure 1). The results also reveal a considerable variability between biological replicates in all three investigated plant species.

### 2.2. Amount and Diversity of Cultivable Bacteria in Lepidium sativum, Cichorium endivia, and Thymus vulgaris

We showed that bacterial species and genera, which contain potential human probiotic acting strains, are present in the phyllosphere of fresh consumed plants and that their abundance is relatively high in all three selected plant species. Now, the question to be answered was, whether these three selected plant species, originating from different plant families, harbor a similar microbiota composition or differ significantly. Therefore, total cultivable bacteria were quantified on a complex nutrient medium and their diversity was estimated using a conventional morphotyping approach. Cultivable bacteria were considerable higher in *L. sativum* with up to 10^8^ CFU g^−1^ fresh weight as compared to *C. endivia* and *T. vulgaris* with up to 10^5^ CFU g^−1^ fresh weight with a high variability in *C. endivia* in the first experiment (Figure 2).

Besides the different amounts of cultivable bacteria, also their morphotyping-based diversity revealed pronounced differences. The high bacterial number of *L. sativum* community, compiled of 28 different morphotypes on standard nutrient agar plates, was shown to be very uneven distributed between the different morphological colony types. That resulted in a lower Shannon diversity index of only 1.95 compared to *C. endivia* with 2.87 and *T. vulgaris* with 2.72 (Table 1).

### 2.3. Isolation and Characterization of Phyllosphere Lactic Acid Bacteria (LAB)

We next attempted to isolate LAB, such as lactobacilli, and bifidobacteria from the phyllosphere of edibles (*C. endivia*, *T. vulgaris, Trigonella foenum-graecum, Coriandrum sativum*, and *Petroselinum crispum*). These bacteria are anaerobic or microaerophilic and many species do not grow well on the surface of solid media incubated aerobically. Thus, these isolates were cultured under microaerophilic conditions using a dedicated medium for LAB. A total of 155 bacterial isolates were obtained: 42 from *T. foenum-graecum*, 33 from *C. endivia*, 27 from *C. sativum*, 18 from *P. crispum*, and 35 from *T. vulgaris*. For screening and identification of those isolates, matrix-assisted laser desorption ionization time-of-flight mass spectrometry (MALDI-TOF MS) was applied. Resulting mass spectra were then matched to a bacterial reference database for identification. The isolates could be divided into four major groups based on the similarity of the respective acquired mass spectra (Figure 3). One major group was formed of isolates identified as *Pediococcus pentosaceus*, the second group consisted of *Enterococcus faecium* isolates, and the two remaining groups contained *Bacillus* species as well as isolates with no reliable identifications. Only 15% of the isolates were considered as not identified, indicating that the reference database allowed identification at least to the genus level of most isolates (Appendix A). However, the best score obtained for identification was 2.493 and the maximal score of 3.000 was never reached, demonstrating the current lack of reference spectra from plant-associated samples.

The bacterial communities found in the phyllosphere of edibles differed with respect to their taxonomic composition. *T. foenum-graecum* leaves were harboring *Bacillus* sp., *E. faecium*, and one *Leuconostoc lactis* strain; *C. endivia* leaves contained *E. faecium* and *P. pentosaceus*; the phyllosphere of *C. sativum* was harboring *P. pentosaceus*, *Staphylococcus* sp. and one *Lactobacillus plantarum*; leaves of *P. crispum* contained *Bacillus* sp. and *P. pentosaceus*; and *T. vulgaris* leaves were colonized by *Bacillus* sp., *P. pentosaceus*, and *Staphylococcus* sp.

Interestingly, the mass spectra of isolates from the same species differed in their protein pattern, depending on the host plant. An example is shown in Appendix A for isolates identified as *P. pentosaceus*. This indicates that specific strains of *P. pentosaceus* colonize specific host plants and that MALDI-TOF-based biotyping is capable of differentiate between those strains.

A typical application of this biotyping method is the dereplication of recurrent isolated microorganisms [41]. However, we found only few isolates with identical mass spectra in our culture collection (Figure 3), demonstrating a high level of biodiversity in the cultivable communities, even if they were obtained on a highly selective medium under microaerophilic conditions.

### 2.4. Phyllosphere Microbiome Composition of Lepidium sativum 

The *L. sativum* phyllosphere bacterial community was investigated using PhyloChip G3 technology in two consecutive independent greenhouse experiments. The aim was to get insight into the entire bacterial community composition and taxa richness of one of the fresh consumed edible plants known for their health promoting effects in human diet. The bacterial genus richness ranged from 49 to 119, belonging to 61 identified and 58 unclassified genera, 63 families, and 17 phyla (Figure 4). The most prominent phyla were Proteobacteria (47 genera) and Firmicutes (21). Other detected phyla were Actinobacteria (9), Cyanobacteria (9), Bacteroidetes (7 genera), Chloroflexi (6), Tenericutes (5), Acidobacteria (3), Planctomycetes (3), Verrucomicrobia (2), and with only one genus: Chlorobi, Gemmatimonadetes, Nitrospirae, Saccharibacteria, (=TM7), Spirochaetes, and the candidate phyla BRC1 and OP11. The presence-absence pattern of the eight biological phyllosphere samples represents a quite high sample to sample variability in the bacterial community composition, although both individual experiments form distinct clusters, except one outlying sample (Appendix A). This high variability between biological replicates was also detected earlier in the quantification of bifidobacteria via qPCR (see Figure 1).

Within the highly diverse and variable community, hybridization signals were identified for the genera *Bifidobacterium*, *Lactobacillus*, and *Streptococcus* (genus-specific OTUs: 111, 160, and 197), that confirm the abundance of species known as probiotic acting bacteria, like two detected strains of *Streptococcus thermophilus* (Table 2). Those potential probiotic strains are marked in Figure 4. Additionally, a high proportion of hybridization signals revealed mostly unclassified taxa from various genera (e.g., less probiotic strains containing Akkermansia, Bacillus, Clostridium, Propionibacterium, and Staphylococcus), families or classes. In particular, Staphylococcus was represented by two potentially skin-probiotic strains of *S. epidermidis*, among others.

## 3. Discussion

A healthy diet is usually closely linked with the recommendation to eat raw vegetables, herbs, or sprouts [42]. Since Pharao’s time, herbs, and fresh produce are known for their healing value [43]. It is assumed that these products are especially healthy to humans because of their high content in vitamins, secondary metabolites, essential oils, or specific mineral compositions. Nowadays, different herb-specific ingredients are known, for example, to suppress harmful microorganisms [44]. Overlooked so far is the additional beneficial effect on the improved balance of intestinal bacterial flora by bacterial communities inhabiting the highly diverse herbs and fresh produce, that probably also includes bacterial species already known as probiotics. Probiotics have recently been defined as “live microorganisms which when administered in adequate amounts confer a health benefit on the host” [45]. Likewise, prebiotics, are non-digestible food fiber components that contribute to host health by activating proliferation and function of beneficial intestinal bacteria [46]. Synbiotics describe a combination of probiotics and prebiotics [23,47] and has been established as medical terms. Probiotic bacteria are processed and enriched for example in yogurt drinks, lactobacillary beverages or other fermented foods and have been medically recognized. Strongest evidence for beneficial probiotic effects has been shown for *Lactobacillus rhamnosus* GG and *Bifidobacterium lactis* BB-12 in curing diarrhea mainly caused by rotaviruses in children (FAO/WHO 2002). Recent publications indicate a much broader range of novel probiotic bacteria, such as *Weissella cibaria* [26].

The presented results show that plant phyllosphere microbiota naturally harbor potential probiotic bacteria, as summarized in Table 3. The well acknowledged, beneficially acting probiotics, containing genera *Bifidobacterium* and *Lactobacillus*, were detected in culture independent molecular investigations. Their population reached ~10^6^ bifidobacteria-specific gene copy numbers per gram plant material which could be translated into ~10^5^ CFU g^−1^ plant fresh weight, assuming a mean 16S rDNA operon copy number per cell [48]. Such a number is comparable to the magnitude as probiotic bacteria *Latobacillus* spp. or *Bifidobacterium longum* and *Streptococcus thermophilus* were administered in specific enriched functional food [49]. In these products it is even difficult to keep living cell numbers above 10^6^ CFU g^−1^ of product for up to 21 days. Often the viable cell counts decrease earlier when foods are kept without adding prebiotic material [49]. Probiotic cells easily survive in plant material since they live in their native habitat and will be digested together with their prebiotic acting plant matrix. Additionally, the investigated plant species revealed highly diverse bacterial communities in their phyllospheres. To which extend these plant inhabiting bacteria survive the acidic stomach environment during the digestion pathway, is still unclear. However, since these bacteria at least partly reside intracellularly of fiber rich material, it can be assumed that they are protected by their plant hosts during this very harsh treatment in the human intestinal tract. Such plant-based transport systems have to be investigated before and probiotics enriched plant products may expand the functional food list.

The constitution of phyllosphere microbiota is linked on the one hand to the developmental stage of the plant [13], but is also influenced by environmental conditions [15]. The presented data indicated a considerable variation between independent experiments, e.g., see Figure 1 for *L. sativum*, or between plant replicate samples within an experiment, e.g., see Figure 2 for *C. endivia*. A certain deviation within the latter has been shown before for raw produce microbiota and is not unexpected [10,15]. The range of variation between independent experiments, especially for *L. sativum*, could be explained by the relatively short time of cultivation. While *T. vulgaris* was cultivated for nine weeks, *L. sativum* was grown only for two weeks until plants were harvested at a stage where they are typically consumed. Hence, it could be possible that a stable microbial community is more likely to be found in plant species that are able to shape their microbiota over a longer period of time, as compared to plant species with a shorter production time. This hypothesis should be tested in analyzing a wider range of independently performed experiments using raw eaten produce with different cultivation regimes.

The aim of culturing phyllosphere bacteria on medium designed for LAB was to analyse whether potential probiotic strains could be isolated from raw eaten herbs and vegetables for future functional studies. From the cultivated isolates, one *L. plantarum* and one *Leuconostoc lactis* isolate were present, both being recognized as probiotic bacteria [55,57,58,72,85]. Enterococci are LAB that are naturally associated with the gastrointestinal tract of humans and animals, but plant-associated *E. faecium* isolates have been found previously [64,86]. Several *E. faecium* strains possess strong probiotic activity [65,67], and since two of the five investigated phyllospheres harbored this species, a possible influence on the intestinal flora of the consumer is assumable. *P. pentosaceus* also belongs to the *Lactobacillaceae*. Its probiotic activity is documented [70] and bacteriocins produced by its strains are able to inhibit the growth of pathogens like *Listeria monocytogenes* [68,69]. Pediococci have been found residing in or on plants [71,87] and the occurrence of this species in phyllosphere samples of four plant species points to a wide-spread distribution with an extensive host range. Previous findings of *S. hominis* in human breast milk, similarly to *E. faecium* and *P. pentosaceus* [83], and its location in two plant phyllospheres point to a probiotic function of this species, although *S. hominis* does not belong to the LAB group and it has not been reported as plant-associated before. Another species of *Staphylococcus*, *S. epidermidis*, found in two plant species, was previously reported as most abundant *Staphylococcus* strain within the human skin microbiome and as a skin probiotic against *Cutibacterium acnes*, even if applied as encapsulated product to avoid bloodstream infections due to its opportunistic pathogenic activity [78,79]. Also the last major group of bacteria identified in this study for three plant hosts, *Bacillus* spp., contains species and strains with probiotic activity, such as *B. coagulans* [55,88]. Surprisingly, no bifidobacterial strains could be cultured although they were detected in two culture independent techniques—the PhyloChip and real-time qPCR. Therefore, future studies should apply newly developed plant-based cultivation media that promise a better chance of cultivating endophytic, plant adapted bacterial strains [89].

Major research efforts are directed to the exploration of plant microbiota with the aim to improve crop performance [1]. However, the functionality and application of these microbiota with respect to human nutrition and well-being is largely untapped. The fact that some novel probiotic bacterial strains appear as opportunistic pathogens on the one hand, but also share antipathogenic properties, even against closely related strains [90], demonstrates the complexity of microbial interactions. This study shows that the phyllosphere of edibles is a substantial source of potential probiotic bacteria, albeit strain- and disease-specific probiotic activity of the isolates has to be resolved in detail in the future. Thus, next studies should focus on investigating the isolated strains functionally, on how these communities can be enriched in the phyllosphere by means of plant production and whether they can be stably transmitted via seeds.

## 4. Materials and Methods

### 4.1. Plant Species and Cultivation

The plant species *Lepidium sativum, Cichorium endivia* var. *crispum*, *Thymus vulgaris*, *Petroselinum crispum*, *Coriandrum sativum*, and *Trigonella foenum-graecum* were grown in two replicated experiments under greenhouse conditions: 16 °C and 21 °C night and day temperature and 60% relative air humidity under natural light during April to July. The greenhouse location was 52°21′16″ N 13°18′22″ E. 

*L. sativum* and *T. foenum-graecum* seeds (2 g seeds per tray) were sawn directly into growth trays filled with Perligran G (Kanuf Perlite GmbH, Dortmund, Germany) and covered five biological replicates. *L. sativum* was grown for two weeks. All other plant species were germinated in quartz sand (≤2 mm), seedlings were transplanted into pots (12 cm in diameter, filled with Fruhstorfer substrate Type P, Germany) and randomly set on trivets to avoid water mediated microbial transfer.

*C. endivia*, one plant per pot, ten pots per biological replicate, with four biological replicates were grown for seven weeks. Ten seedlings of *T. vulgaris* were transplanted into one pot; three pots per biological replicate were established in four biological replicates. *T. vulgaris* plants were grown for nine weeks. Four seedlings of *P. crispum* and *C. sativum* were transplanted per pot, five pots per biological replicate and four biological replicates. Both plant species were grown for six weeks. Above ground plant material was aseptically collected in a growth stage which is representative for usual consumption.

### 4.2. Determination of Bifidobacterial-Specific Gene Copy Numbers Using Quantitative Real-Time PCR

To demonstrate the presence of potential probiotic acting bacteria in leaf material, the well-known human probiotic acting bacterial genus Bifidobacterium was used as model. One microliter (concentration 50 ng µL^−1^) of total DNA samples was extracted from lyophilized plant tissue using DNeasy Plant MiniKit (Qiagen) was investigated in a bifidobacteria genus-specific qPCR. Therefore, primers g-Bifid-F (5′-CTCCTGGAAACGGGTGG-3′) and g-Bifid-R (5′-GGTGTTCTTCCCGATATCTACA-3′) were used [40]. Quantitative real-time PCR (qPCR) was performed with iQ^TM^ SYBR Green Supermix (Biorad, Hercules, CA, USA), the amplification protocol of five minutes initiation step at 94 °C and 30 cycles of 20 s at 94 °C, 20 s 55 °C, and 30 s 72 °C followed by 5 min 72 °C reassociation and a melting curve incrementing 0.5 °C in 85 cycles starting at 55 °C to verify the product purity and specificity. The DNA amplified in technical triplicates was used in the original concentration, which was isolated from plant material and in a dilution 1:10. The reaction was performed in a CFX96 Touch real-time PCR detection system (Biorad, USA). A standard dilution series of a purified PCR product of *Bifidobacterium breve* (DSM 20213) was performed to calculate copy numbers according to a regression curve. Copy numbers were calculated via this regression curve.

### 4.3. Bacterial Cultivation

#### 4.3.1. Total Cultivable Bacterial Numbers and Diversity on Complex Nutrient Agar

Fresh plant material (5 g) was ground, 1:10 diluted in 0.05 M NaCl buffer solution, and shaken on a horizontal shaker (180 rpm) with 6 glass beads (5 mm diameter) at 4 °C for one hour. A serial dilution (5 times 1:10 diluted) was prepared, 100 µL of each dilution were spread petri dishes of standard nutrient agar (Carl Roth GmbH, Karlsruhe, Germany) (three replicates per dilution and sample) and incubated at 37 °C for 72 h. Colony forming units (CFUs) were counted and classified according their morphology (color, side face, circumference, surface, diameter, and transparency). Different morphotypes and numbers were used to calculate the Shannon diversity index (Shannon 1948) H = −∑(pi*lnpi), and the evenness of morphotype occurrence E = H/H_max_, where pi = ni × N^−1^, N = the total number of CFUs, ni = number CFUs per colony type, S = number of different colony types, H_max_ = lnS.

#### 4.3.2. Specific Enrichment for Bifidobacteria and Lactobacilli

MRS-Agar (deMan-Rogosa-Sharpe-Agar, MERCK, Germany), which favors the growth of lactic acid bacteria, was used to cultivate LABs, and TOS-Propionate agar with lithium mupirocin (MERCK, Germany) was applied to cultivate bifidobacteria. Bacterial washes from phyllosphere plant samples (as described in total cultivable bacterial numbers detection part) and their first three diluted samples (10^−1^, 10^−2^ und 10^−3^) were spread in triplicates on media. They were incubated in anaerobic jars (MERCK, Germany) with Oxoid AnaeroGen satchets (Thermo Scientific, Schwerte, Germany) for generating a microaerophilic atmosphere in the jars for 72 h at 37 °C. All emerging colonies were counted and collected for protein pattern composition analysis using the MALDI-TOF biotyper approach. No colonies were detected on TOS-agar.

### 4.4. Protein Extraction from Bacterial Colonies for MALDI-TOF Biotyper Analysis

For the extraction, the bacteria were suspended in 150 μL of MilliQ water and vortexed. Next, 450 μL of 100% ethanol was added, vortexed, and centrifuged (13,000× *g*, 2 min). The supernatant was discarded, pellets were dried at room temperature, resuspended in 5 μL of 70% formic acid and vortexed, then 5 μL of acetonitrile was added to the mix, strongly vortexed till the pellet completely dissolved and centrifuged as above. Supernatants (1 μL) were spotted onto the MALDI-TOF polished steel target plate (Bruker, Bremen, Germany) and dried at room temperature. Subsequently, 0.5 μL of MALDI-TOF matrix (a saturated solution of α-cyano-4-hydroxycinnamic acid (Bruker) in 50% acetonitrile and 2.5% trifluoroacetic acid) was applied onto the colony and allowed to dry before testing. A bacterial test standard (Bruker) was used to calibrate the MALDI-TOF method prior to each run.

### 4.5. MALDI-TOF Mass Spectrometric Measurements and Isolate Identification

Automated acquisition of the mass spectra (2000–20,000 Da) was done using an UltraFlex MALDI-TOF mass spectrometer (Bruker, Germany), equipped with a nitrogen laser, working in linear positive mode and controlled by FlexControl software (v3.4, Bruker). Spectra were inspected using the FlexAnalysis software (v3.4, Bruker). Processing of raw spectra, search against the bacterial database (7,014 entries of reference strains) and similarity clustering was performed using MALDI BioTyper software (v3.1, Bruker). For bacterial identification, a log score is generated by the software based on the similarity to database entries, ranging from 0.00 to 3.00. A score of 2.30–3.00 was considered as highly probable species identification, a score of 2.00–2.29 as secure genus and probable species identification, and a score of 1.70–1.99 as probable genus identification.

### 4.6. Plant Microbiome Analysis

For the comprehensive microbiome composition of above ground usually fresh indigested herbs, *Lepidium sativum* leaves were investigated applying the PhyloChip Array method (PhyloChip G3, Second Genome, San Bruno, CA, USA). Plant material was freeze-dried. Total DNA was extracted using DNeasy Plant MiniKit (Qiagen, Germany) and moved forward for hybridization. After adding PhyloChip Control Mix^TM^ to each amplified product, these products were fragmented, biotin labeled and hybridized to the PhyloChip^TM^ Array. Data analysis was performed using PhyCA-Stats^TM^ analysis software (Second Genome, USA). Significant present genus-specific OTUs per sample were summarized in an OTU table. The absence-presence pattern of the genera were illustrated as heatmap exerting the *heatmap.2()* function of the R package *ggplot2*. The map was further arranged according the genera absence-presence pattern similarity, applying the binary method of the *dist()* function in R on rows, and according the sample specific UPGMA cluster (columns). Taxonomic affiliation of the genus-specific OTUs, assigned within the OTU table, were parsed with an in-house script into a Newick tree, where each node/furcation towards the leaf node can be equated with a new taxonomic level, starting from the kingdom Bacteria at root towards the genus OTU at each leaf. The tree was visualized with iTol [72]. Non-allocable taxonomic paths towards the leaves were plotted as spotted lines. The genus OTU identifiers were used as leaf labels and their background colored according its phylum. Their absence (gray box) and presence (black box) across all eight samples, encompassing the two independent experiments with four replicates, is displayed in the surrounding circles. Genera including potential probiotic strains were emphasized with a green dot behind the leaf label.

## Figures and Tables

**Figure 1 ijms-20-03661-f001:**
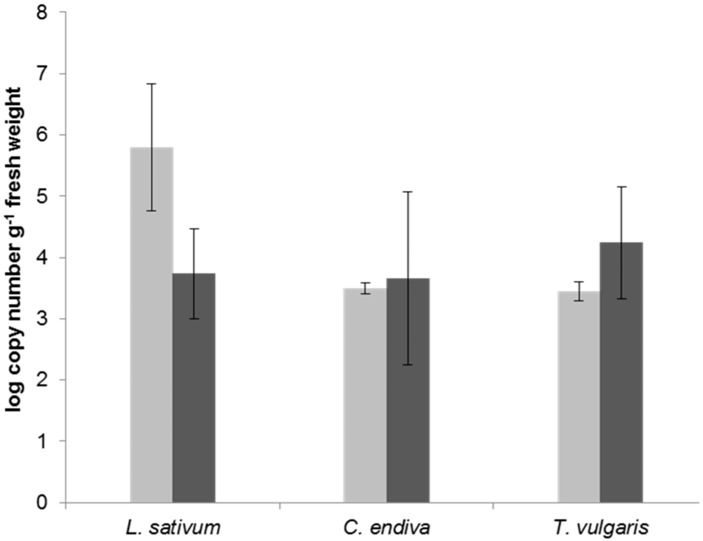
Abundance of bifidobacteria (log_10_ copy number per gram fresh weight) in the phyllosphere of three plant species determined via qPCR. Shown are mean values of five plants for the first (light gray bars) and second (dark gray bars) experiment, with the standard deviation as error bars.

**Figure 2 ijms-20-03661-f002:**
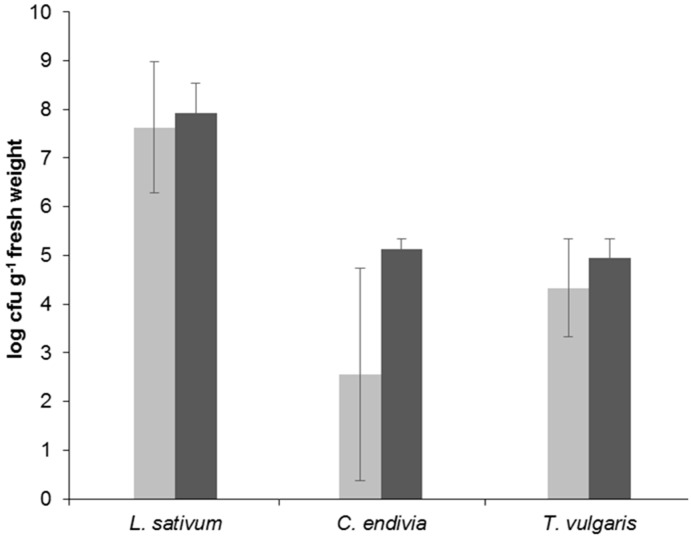
Viable counts (log_10_ CFU g^−1^ fresh weight) of cultivable bacteria obtained from three plant species. Shown are mean values of five plants for the first (light gray bars) and second (dark gray bars) experiment, with the standard deviation as error bars.

**Figure 3 ijms-20-03661-f003:**
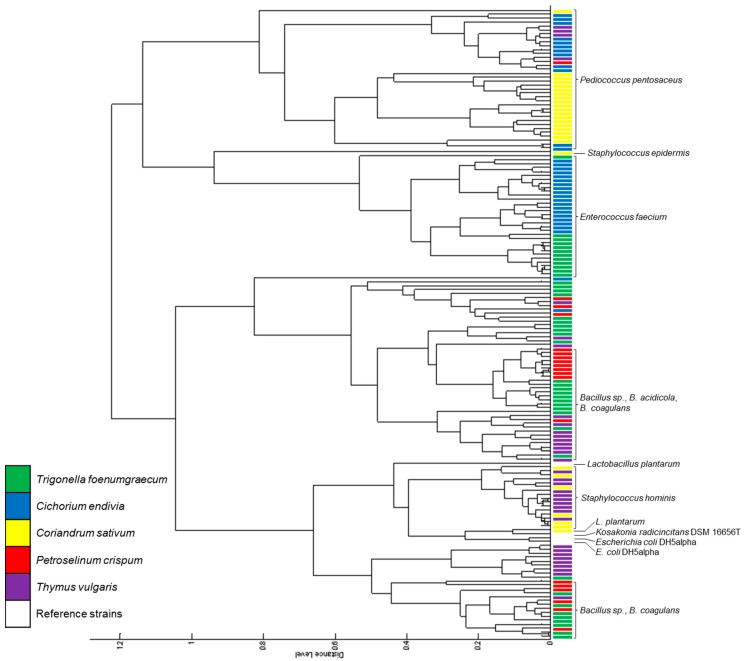
Cluster analysis of bacterial mass spectra obtained from selected reference strains and plant isolates. The latter are labeled in color according to the plant host. The identification of major isolate groups is shown on the right. Distance is displayed in relative units. For the full list of identified isolates, please refer to Appendix A.

**Figure 4 ijms-20-03661-f004:**
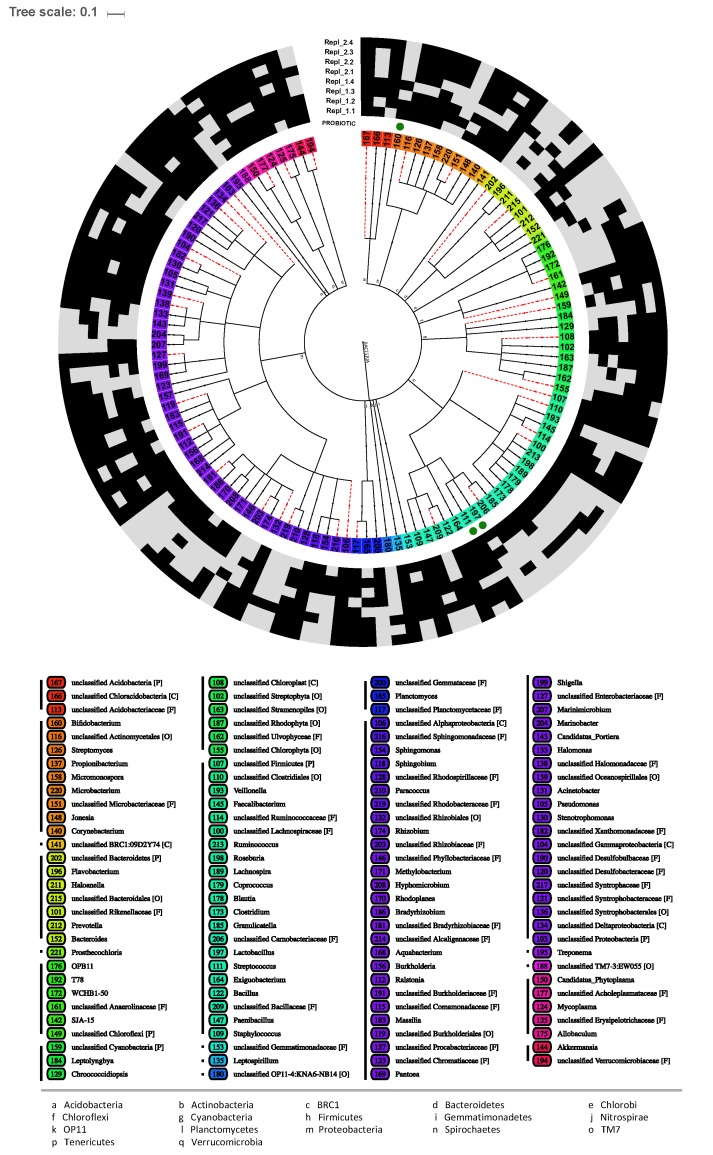
Taxonomic affiliation and absence-presence pattern of *Lepidium sativum* phyllosphere microbiota bacterial genera detected by the PhyloChip G3 technology. Taxonomic assignment of the genus-specific OTUs are displayed on the leaves and colored according their phyla (a–q). Spotted red lines indicate uncertain assignments, whereby their last known taxonomic rank (P: Phyla; C: Class; O: Order; F: Family) is listed behind the respective affiliation in the legend. The surrounding circles, describing the four replicates for individual experiments (Repl_1, Repl_2), reveal the presence (black) and absence (gray) of the appropriate genus. Genera with high probability of containing probiotic species are marked with a green dot.

**Table 1 ijms-20-03661-t001:** Morphotype diversity index (Shannon index) of bacterial strains isolated from three plant species.

Biodiversity Indices	*L. sativum*	*C. endivia*	*T. vulgaris*
**N**	8018	56	276
**S**	28	26	30
**H**	1.95	2.87	2.72
**H_max_**	3.33	3.26	3.40
**E**	0.59	0.88	0.80

N: colony-forming units count; S: number of morphotypes; H: Shannon index calculated based on the distribution of N on S; H_max_: maximal Shannon index; E: eveness.

**Table 2 ijms-20-03661-t002:** Taxonomic affiliation of genera and species, which are known to contain effective probiotic strains, detected in *L. sativum* phyllosphere microbiome via PhyloChip analysis.

PHYLA	ORDER	GENUS	SPECIES
***Actinobacteria***	*Bifidobacteriales*	*Bifidobacteria*	unclassified
***Firmicutes***	*Lactobacillales*	*Lactobacillus*	unclassified
		*Streptococcus*	*thermophilus*
		*Streptococcus*	unclassified

Other genera containing potential probiotic genera found: *Akkermansia*, *Bacillus*, *Clostridium* and *Propionibacterium*, and *Staphylococcus* (e.g., *S. epidermidis*).

**Table 3 ijms-20-03661-t003:** Potential probiotic bacterial species detected in the phyllosphere of raw eaten produce of the present study. Related references demonstrate the verified probiotic character of the strains of the same species.

Probiotic Group	Species	Phyllosphere (Plant)	Probiotic Reference
***Lactobacillus *^1^**	(*Bacteria*; *Terrabacteria group*; *Firmicutes*; *Bacilli*; *Lactobacillales*; *Lactobacillaceae*)		
	*Lactobacillus* sp.	LS	[50,51,52,53,54,55]
	*L. plantarum*	CS	[20,54,55,56,57,58,59]
***Bifidobacteria***	(*Bacteria*; *Terrabacteria group*; *Actinobacteria*; *Actinobacteria*; *Bifidobacteriales*)		
	*Bifidobacteria* sp.	LS	[53,55,60]
***Streptococcus *^1^**	(*Bacteria*; *Terrabacteria group*; *Firmicutes*; *Bacilli*; *Lactobacillales*; *Streptococcaceae*)		
	*Streptococcus* sp.	LS	[20,54,55]
	*S. thermophilus*	LS	[20,53,54,61,62]
***Enterococcus *^1^**	(*Bacteria*; *Terrabacteria group*; *Firmicutes*; *Bacilli*; *Lactobacillales*; *Enterococcaceae*)		
	*E. faecium*	CE, TF	[20,53,54,55,60,63,64,65,66,67]
***Pediococcus *^1^**	(*Bacteria*; *Terrabacteria group*; *Firmicutes*; *Bacilli*; *Lactobacillales*; *Lactobacillaceae*)		
	*P. pentosaceus*	CE, CS, PC, TV	[20,53,55,59,60,68,69,70,71]
***Leuconostoc*^1^**	(*Bacteria*; *Terrabacteria group*; *Firmicutes*; *Bacilli*; *Lactobacillales*; *Leuconostocaceae*)		
	*L. lactis*	TF	[72]
***Bacillus***	(*Bacteria*; *Terrabacteria group*; *Firmicutes*; *Bacilli*; *Bacillales*; *Bacillaceae*)		
	*Bacillus* spp.	LS, PC, TF, TV	[55,73,74]
***Propionibacterium***	(*Bacteria*; *Terrabacteria group*; *Actinobacteria*; *Actinobacteria*; *Propionibacteriales*; *Propionibacteriaceae*)		
	*Propionibacterium* sp.	LS	[20,54,55]
***Akkermansia***	(*Bacteria*; *PVC group*; *Verrucomicrobia*; *Verrucomicrobiae*; *Verrucomicrobiales*; *Akkermansiaceae*)		
	*Akkermansia* sp.	LS	[20,75,76,77]
***Staphylococcus***	(*Bacteria*; *Terrabacteria group*; *Firmicutes; Bacilli*; *Bacillales*; *Staphylococcaceae*)		
	*S. epidermidis*	CS	[78,79,80]
	*S. hominis*	CS, TV	[80,81,82,83]
***Clostridium***	(*Bacteria*; *Terrabacteria group*; *Firmicutes*; *Clostridia*; *Clostridiales*; *Clostridiaceae*)		
	*Clostridium* sp.	LS	[29,84]

^1^ LAB (lactic acid bacteria); CE: *Cichorium endivia* L.; CS: *Coriandrum sativum* L.; LS: *Lepidium sativum* L.; PC: *Petroselinum crispum* L.; TF: *Trigonella foenum-graecum* L.; TV: *Thymus vulgaris* L.

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
