# Peer review of "Culture Dependent and Independent Analysis of Potential Probiotic Bacterial Genera and Species Present in the Phyllosphere of Raw Eaten Produce"

_ijms, 2019, doi:10.3390/ijms20153661_

Round 1

Reviewer 1 Report

Overall the findings from this study shed light on the importance of understanding the abundance and diversity of microbes present in certain plants and crops that could lead to improvements in human health. Currently the use of plant microbiota remains focused on general crop growth and performance while the possible benefits of plant microbes in human diets and general well-being remains severely understudied. This study attempted to provide a glimpse of the diversity and magnitude of microbes in plants specifically looking at the phyllosphere. According to the study, the physllospere of raw eaten edibles contains a highly abundant population of probiotic bacteria however, composition of the microbes is dependent on the host itself. Although this study is descriptive in nature, it is certainly interesting and worthy of publication.

Major Concerns

·         The paper could benefit from improved English grammar

·         Why did the authors choose to focus the study on L. sativum, C. endiva, and T. vulgaris? The reasoning behind choosing these three species specifically was not explained.

Minor Concerns

·         Table 1 presents a lot of information on general probiotic bacteria when only a few specific species are mentioned throughout the rest of the study. While this information provides a lot of insight in the vast abundance of probiotics it may not be necessary or directly relevant to the results of this study.

·         Figure 1 shows the abundance of bifidobacteria in the phyllosphere for two separate experiments but never explains the huge variability in abundance found in L sativum between the two experiments.

·         Line 18: Missing the word “of” prior to the word “those”

·         Line 25: swap the words “probiotic” and “added”

·         Line 48-50: It’s mentioned that conventional vs organic plants were compared in the cited study but I’m left wondering what those differences were. Were both the plant genetics and microbiota different in conventional vs. organic plants?

·         Line 70, 353, 361: large font size on A. muciniphila and hyperlinks

Author Response

Overall the findings from this study shed light on the importance of understanding the abundance and diversity of microbes present in certain plants and crops that could lead to improvements in human health. Currently the use of plant microbiota remains focused on general crop growth and performance while the possible benefits of plant microbes in human diets and general well-being remains severely understudied. This study attempted to provide a glimpse of the diversity and magnitude of microbes in plants specifically looking at the phyllosphere. According to the study, the physllosphere of raw eaten edibles contains a highly abundant population of probiotic bacteria however, composition of the microbes is dependent on the host itself. Although this study is descriptive in nature, it is certainly interesting and worthy of publication.

Major Concerns

The paper could benefit from improved English grammar

Thank you for this suggestion. The manuscript has been edited for grammar.

Why did the authors choose to focus the study on L. sativum, C. endiva, and T. vulgaris? The reasoning behind choosing these three species specifically was not explained.

In order to make our choice of plants more clear, we have added the following sentence to the Introduction: ‘The investigated plant species were selected due to their widespread raw consumption, economic value and their affiliation to different plant orders, assuming that the plant genotype affects the presence of potential probiotic bacteria.’

Minor Concerns

Table 1 presents a lot of information on general probiotic bacteria when only a few specific species are mentioned throughout the rest of the study. While this information provides a lot of insight in the vast abundance of probiotics it may not be necessary or directly relevant to the results of this study.

We completely agree with the reviewer’s remark. Originally, we aimed to give an overview of currently known probiotic species / strains that we were screening for in raw eaten produce. We have removed the table from the introduction, but indicate their importance, and added a smaller version to the discussion section (Table 3), considering only the species found in the respective plant phyllospheres and their references confirming the potential probiotic property.

Figure 1 shows the abundance of bifidobacteria in the phyllosphere for two separate experiments but never explains the huge variability in abundance found in L sativum between the two experiments.

We agree with the reviewer that this point was omitted in the discussion. We have added a paragraph with a possible explanation for the observed experimental variations.

Line 18: Missing the word “of” prior to the word “those”

This has been corrected.

Line 25: swap the words “probiotic” and “added”

This has been corrected.

Line 48-50: It’s mentioned that conventional vs organic plants were compared in the cited study but I’m left wondering what those differences were. Were both the plant genetics and microbiota different in conventional vs. organic plants?

Thank you for this remark. We have added more information with respect to this citation: ‘They also identified differences between conventional and organically farmed produce, showing that organic-labelled spinach, lettuce and tomatoes had a greater OTU richness, but peaches and grapes had a lower OTU richness, when compared to the conventional produce type.’

Line 70, 353, 361: large font size on A. muciniphila and hyperlinks

This has been corrected.

Reviewer 2 Report

The aim of the manuscript was to report on the probiotic microbiota present in the phyllosphere or raw eaten produce. The methodology applied was suitable for both culture -dependent and -independent assessment and the manuscript is overall very well written. However, there is one major concern and a few minor suggestions.

The major concern refers to the use of the term ‘probiotic’. It has been exhibited by numerous studies that such a property is strain dependent and requires a lot of in vitro and in situ experimentation. The term putative/potential probiotic usually refers to strains that have exhibited their probiotic potential in vitro and in situ experiments are yet to be performed. In this manuscript, probiotic capacity/action is inferred only by the genus or species name, which is incorrect. If the authors want to use any of these terms, they should present the results of the respective experiments.

Minor issues:

l. 17-18 it should read ‘...in the phyllosphere of those...’

l. 18 and throughout the text: CFU is an acronym and should be written in capital letters.

l. 63-65. these lines suggest that LAB and bifidobacteria are probiotics. The strain-dependent property of this character should be emphasized in these lines as well.

Table 1 would fit nicely in a review article; it can be omitted.

l. 319-320. MRS do not suppress growth of competitors, the incubation conditions do. Please update

Author Response

The aim of the manuscript was to report on the probiotic microbiota present in the phyllosphere or raw eaten produce. The methodology applied was suitable for both culture -dependent and -independent assessment and the manuscript is overall very well written. However, there is one major concern and a few minor suggestions.

The major concern refers to the use of the term ‘probiotic’. It has been exhibited by numerous studies that such a property is strain dependent and requires a lot of in vitro and in situ experimentation. The term putative/potential probiotic usually refers to strains that have exhibited their probiotic potential in vitro and in situ experiments are yet to be performed. In this manuscript, probiotic capacity/action is inferred only by the genus or species name, which is incorrect. If the authors want to use any of these terms, they should present the results of the respective experiments.

We agree with that concern, and tried to emphasize that further studies on isolates strain- and disease-specific probiotic activities has to be clarified in the next studies. We also changed the writing in the manuscript to “genera and species containing potential probiotic acting strains”. So we will not state here that our isolates are acting probiotics, actually.

Minor issues:

l. 17-18 it should read ‘...in the phyllosphere of those...’

This has been corrected.

l. 18 and throughout the text: CFU is an acronym and should be written in capital letters.

Thank you for this advice, this has been corrected

l. 63-65. these lines suggest that LAB and bifidobacteria are probiotics. The strain-dependent property of this character should be emphasized in these lines as well.

Thank you for that valuable hint. Now we point to the strains- and disease-specificity of probiotic bacteria, as described by McFarland et al. 2018, see Introduction.

Table 1 would fit nicely in a review article; it can be omitted.

We are in compliance with the reviewer’s comment on Table 1 and have delete the table from the introduction. Further we have added a tiny version of it to the discussion section (Table 3) connecting the detected bacteria of the plants phyllosphere to their references, describing their probiotic properties.

l. 319-320. MRS do not suppress growth of competitors, the incubation conditions do. Please update

Thanks for the correction. We changed the description and added details to the incubation condition.

Reviewer 3 Report

In this study, the author selected Lepidium sativum, Cichorium endivia and Thymus vulgaris as the main research object and protagonist. However, there is a lack of representative analysis and explanation from the cultivation latitude area, consumption quantity and mode, and chloroplast characteristics. The author should at least explain these aspects and decision making points.

The gene copy numbers were so important in this study. Author only mention on some analysis method but no to point out the important level. Please explain in the text about the copy number and the importance and key points in this study so that the reader can understand and catch your points.

Before the MALDI-TOF mass spectrometric measurements, the selected clones cultured method very important. Different media, different culture temperature and time would get different results. Specially, in this study author used the data to do more comparisons and analysis. in the related paragraph and Line 318-325 should be description more details and clear about the conditions of the culture stags.

MALDI-TOF mass spectrometric is used for measurements and isolate identification, but most of the MALDI-TOF biotyper databases are pathogens, and are from clinical strains, rarely lactic acid bacteria or probiotics, especially from environmental or agricultural strains. . The author is asked to explain the source of the strain and the source of the strain, and the method of identification when the strain is identified..

MALDI-TOF mass spectrometry analyzes strains based on more than five scores and colors. The author does not explain how score is generated. How is it graded in this study? What is the basis for grading?

In Figure 1 and Figure 2, only 3 plants were used as the basis for the analysis. However, the design of the control group or the control group is missing. Please explain how the results are verified. Was it a cross-comparative or contrasting principle used?

Figure 3 was the result of the Cluster analysis, but it did not indicate what data were analyzed. What kind of proteins were of the strain? Was it the group? Nor did it provide or describe the main parameters of the Cluster analysis comparison? In Supplemental Figure 1 MALDI this information was missing in the -TOF MS profiles?

Figure 4. Taxonomic tree was missing the parameters of the UPGMA clustering algorithm?

Very importantly, the functionality of probiotic bacteria is based on “Strain” identification rather than “Species” identification. Therefore, only the Species analysis or identification, there is no specific and substantial significance in the function of probiotic bacteria.

Author Response

In this study, the author selected Lepidium sativum, Cichorium endivia and Thymus vulgaris as the main research object and protagonist. However, there is a lack of representative analysis and explanation from the cultivation latitude area, consumption quantity and mode, and chloroplast characteristics. The author should at least explain these aspects and decision making points.

Thank you for this remark. We have tried to address this comment by adding information to the Introduction to explain the choice of plant species, as well as to the Materials and Methods section to give information on the location of plant cultivation. We have not analysed chloroplast characteristics, so unfortunately, we cannot comment on this.  

The gene copy numbers were so important in this study. Author only mention on some analysis method but no to point out the important level. Please explain in the text about the copy number and the importance and key points in this study so that the reader can understand and catch your points.

We added in the discussion part the meaning of gene copy numbers. The translation of gene copy number into bacterial cell number can be calculated by dividing gene copy number by about 4 (as a mean value) which leads to an approximate bacterial cell number since bacterial genoms contain between 2 and 7 16S rDNA gene copies.

Before the MALDI-TOF mass spectrometric measurements, the selected clones cultured method very important. Different media, different culture temperature and time would get different results. Specially, in this study author used the data to do more comparisons and analysis. in the related paragraph and Line 318-325 should be description more details and clear about the conditions of the culture stags.

Thank you for this comment; we agree that the methods need to be explained in detail. We have added missing information to the respective part.

MALDI-TOF mass spectrometric is used for measurements and isolate identification, but most of the MALDI-TOF biotyper databases are pathogens, and are from clinical strains, rarely lactic acid bacteria or probiotics, especially from environmental or agricultural strains. . The author is asked to explain the source of the strain and the source of the strain, and the method of identification when the strain is identified..

We agree with the reviewer that the current available databases for the identification of microbes are designed specifically for clinical isolates. Nevertheless, there are numerous lactic acid bacteria present as database entries, as seen in our manuscript. Here, we were able to identify 85 % our isolates at least to the genus level. To our knowledge, the source of the strains included in the biotyper database is from culture collections, such as the ‘German Collection of Microorganisms and Cell Cultures (DSMZ)’. Here, strain genomes are sequenced completely to ensure secure identification. Then, those strains are provided to Bruker company and there the intact protein mass spectra are acquired that are needed for generating the database entry. In order to identify an unknown isolate, the user acquires the intact protein mass spectrum and a software compares this data with a database entry. A score is generated, ranging from 0.00 to 3.00, that indicates the level of similarity between the unknown isolate and the database entry. 

MALDI-TOF mass spectrometry analyzes strains based on more than five scores and colors. The author does not explain how score is generated. How is it graded in this study? What is the basis for grading?

The score representing the output of the comparison between the unknown isolate and the database entry is calculated based on a pattern recognition algorithm that uses peak positions, peak intensity distributions as well as peak frequencies to find the best match. According to Bruker company the following principle is followed to calculate the score in the Biotyper software:

‘The first step of data processing is converting raw mass spectra into peak lists. In a second step, these peak lists are compared (matched) with a specified subset of patterns in the reference database.

The matching algorithm computes three separate values for three fundamental characteristics of the sample and the reference spectra.

First, the number of signals      in the reference spectrum that have a closely matching partner in the      unknown spectrum are calculated. No matches returns a value = 0 and a      complete match returns a value = 1.

Then, the number of signals      in the unknown spectrum that have a closely matching partner in the      reference spectrum are determined. No matches returns a value = 0 and a      complete match returns a value = 1.

Finally, the symmetry of the      matching signal pairs is computed. If the high-intensity signals of the      unknown spectrum correspond with the high-intensity signals of the      reference spectrum and the low-intensity signals also correspond, this      results in high symmetry value and the so-called correlation matrix yields      a value close to 1. If the matching pairs of signals show no symmetry at      all, this results in a value close to 0.

These three values are multiplied together and the result is normalized to 1000. The score value is the log of this result. The maximum obtainable score value is 3 (= log 1000).

Subsequent ranking of the score values puts the best correlation at the top of the list. The higher the score value, the more probable the classification of the species. A color-coded display of the score value enables rapid evaluation of the results.

Score values ≥ 2.0 can be considered as a probable classification.’

According to the Biotyper software, the score is used in the following way:  ‘A score of 2.30–3.00 was considered as highly probable species identification, a score of 2.00–2.29 as secure genus and probable species identification and a score of 1.70–1.99 as probable genus identification.’ and this is stated in the Materials and Methods section.

In Figure 1 and Figure 2, only 3 plants were used as the basis for the analysis. However, the design of the control group or the control group is missing. Please explain how the results are verified. Was it a cross-comparative or contrasting principle used?

This is a misunderstanding. We have analysed for L. sativum two g seeds per replicate and five biological replicates, for C. endivia 10 plants per replicate and four biological replicates and for T. vulgaris three plants per replicate and four biological replicates. This is stated in the Materials and Methods section. Since we did not treat the plants in any way, we did not see the need of a control group. All plants grown were considered as controls.  

Figure 3 was the result of the Cluster analysis, but it did not indicate what data were analyzed. What kind of proteins were of the strain? Was it the group? Nor did it provide or describe the main parameters of the Cluster analysis comparison? In Supplemental Figure 1 MALDI this information was missing in the -TOF MS profiles?

The hierarchical clustering of protein pattern is a feature of the Biotyper software. This is managed by an external MATLAB software tool that is integrated into MALDI Biotyper. As stated in the Results section, mass spectra of the respective isolated strains were used to create the dendrogram. The nature of proteins present in the mass spectra and used for similarity searches is not known and it is also not necessary to know the identity of measured proteins. It is rather used as a molecular fingerprint because this mass spectrum is species-specific for a large number of microorganisms.

The purpose of Supplemental Figure 1 is to show the variability of isolates all assigned to Pediococcus pentosaceus that have been isolated from different host plants. When compared to Figure 3, it can be seen that the strain isolated from C. sativum is located in a different cluster as the other strains.

Figure 4. Taxonomic tree was missing the parameters of the UPGMA clustering algorithm?

Indeed, the term “tree” was misleading in the figure title and suggested any sequence based clustering algorithm behind. Here we present only a taxonomic affiliation in a hierarchical manner, as described in the method section under “Plant microbiome analysis“. As we used only the taxonomic paths given as result from the PhyloChip G3 analysis of Second Genome, we have altered the Figure title to: “Taxonomic affiliation …”

Very importantly, the functionality of probiotic bacteria is based on “Strain” identification rather than “Species” identification. Therefore, only the Species analysis or identification, there is no specific and substantial significance in the function of probiotic bacteria.

Thank you for pointing on that issue. In the Discussion we now indicate that the isolates strain- and disease-specific probiotic activity has to be investigated in future studies.

We also adjusted the title of our paper to reveal that we only searched for species and genera which are known to contain already proved probiotic strains.

Round 2

Reviewer 2 Report

the authors have addressed all issues

Reviewer 3 Report

In the Materials and Methods section, please correct "ul" to "uL" and so on.